# Seroprevalence of viral and vector-borne bacterial pathogens in domestic dogs (*Canis familiaris)* in northern Botswana

Riley Thompson[1], Hayley Adams[2], Agricola Odoi[3], Melissa Kennedy[3]*

**1** Department of Large Animal Clinical Sciences, College of Veterinary Medicine, University of Tennessee, Knoxville, TN, United States of America, **2** Silent Heroes Foundation, Saint Simon Island, GA, United States of America, **3** Department of Biomedical and Diagnostic Sciences, College of Veterinary Medicine, University of Tennessee, Knoxville, TN, United States of America

* mkenned2@utk.edu

## Abstract

### Background

Domestic dogs (*Canis familiaris*) have the potential to act as disease reservoirs for wildlife and are important sentinels for common circulating pathogens. Therefore, the infectious disease seroprevalence among domestic dogs in northern Botswana may be indicative of pathogen exposure of various wildlife species. The objective of this study was to assess the seroprevalence of *Ehrlichia spp.*, *Borrelia burgdorferi*, *Anaplasma spp.*, *Dirofilaria immitis*, canine adenovirus, canine parvovirus, and canine distemper virus in domestic dogs as proxies of disease prevalence in the local wildlife in the Okavango Delta region of Botswana. Statistical analysis assessed crude and factor-specific seroprevalence proportions in relation to age, sex, and geographical location as predictors of seropositivity. Logistic regression was used to identify adjusted predictors of seropositivity for each of the pathogens of interest.

### Results

Samples from 233 dogs in a total of seven locations in Maun, Botswana, and surrounding villages were collected and serologically analyzed. No dogs were seropositive for *B. burgdorferi*, while low seroprevalence proportions were observed for *Anaplasma spp.* (2.2%) and *D. immitis* (0.9%). Higher seroprevalence proportions were observed for the tick-borne pathogen *Ehrlichia spp.* (21.0%), and 19.7% were seropositive for canine adenovirus (hepatitis). The highest seroprevalence proportions were for canine parvovirus (70.0%) and canine distemper virus (44.8%). The predictors of seropositivity revealed that adults were more likely to be seropositive for canine adenovirus, canine distemper virus, and canine parvovirus than juveniles, and location was a risk factor for canine adenovirus, canine distemper virus, canine parvovirus, and *Ehrlichia spp*.

### Conclusions

Results indicate that increasing tick control and vaccination campaigns for domestic dogs may improve the health of domestic animals, and potentially wildlife and humans in the

**Funding:** Funded by MAK. Grant #D16CA-605. Morris Animal Foundation Veterinary Student Scholars. https://www.morrisanimalfoundation. org/sites/default/files/filesync/VSS-Guidelines.pdf. The funders had no role in study design, data collection and analysis, decision to publish, or preparation of the manuscript.

**Competing interests:** The authors have declared that no competing interests exist.

Okavango Delta since viral and vector-borne bacterial pathogens can be transmitted between them.

## Introduction

Vaccination of domestic dogs has been reported as a method of wildlife conservation [1] with the implication that prevalence of transmissible diseases in the domestic canine population has the potential to affect disease burden in wildlife, including both wild felids and wild canids. African wild dogs (*Lycaon pictus*), a wild canid species in sub-Saharan Africa, are endangered according to the International Union for the Conservation of Nature Redlist [2], and the black-footed cat (*Felis nigripes*), cheetah (*Acinonyx jubatus*), and lion (*Panthera leo*) are all vulnerable species in Botswana [2] that can be negatively impacted by domestic dog viral and vector-borne bacterial pathogens [3, 4, 5]. Capturing a sufficient number of African wild dogs, black-footed cats, cheetah, or lion to perform seroprevalence surveys is not always feasible due to the risk associated with anesthesia necessary to collect blood samples from these animals. This is particularly concerning due to the low numbers of individuals as indicated by their conservation status. McRee, *et al.* [6] performed a prevalence evaluation of viral pathogens in domestic dogs in northwest Zimbabwe as a representation of wildlife viral disease prevalence in the region, particularly African wild dogs.

*Ehrlichia spp.*, *Anaplasma spp.*, and *Borrelia burgdorferi* (Lyme disease) are bacterial pathogens that are transmitted by tick bites [7, 8, 9]. *Dirofilaria immitis*, or heartworm disease, is a blood-borne parasite transmitted by mosquito bites [10], and canine distemper (CDV), parvovirus (CPV), and adenovirus (CAV) are viral diseases transmitted between individuals [11, 12, 13]. All of these common pathogens in dogs can cause significant morbidity and mortality. While the viral diseases, Lyme disease, and heartworm disease can be prevented either by vaccination or monthly heartworm preventative medication, many communities in southern Africa do not have the resources to pay for these medications for their animals. Thus, these preventable diseases may be widespread.

Botswana is a land-locked country in southern Africa and is home to the Okavango Delta, a diverse wetland habitat. Not only is the Okavango Delta home to countless species, it is the center for tourism in the country, which has become the second most important industry in Botswana after diamond mining [14]. The Okavango Delta of Botswana is rich with wildlife which have the chance to interact with domestic animal populations. This potentially results in cross-species transmission of infections between domestic and wild animals implying that infectious disease exposure in domestic animals might mirror those of wildlife. As disease prevalence of common infectious diseases in wild carnivores is unknown in the Okavango Delta, this presents the opportunity to use domestic dogs as sentinels for infectious disease exposure in wildlife. Therefore, the objective of this study was to assess the seroprevalence of common infectious diseases (*Ehrlichia spp.*, *Borrelia burgdorferi*, *Anaplasma spp.*, *Dirofilaria immitis*, CDV, CAV, and CPV) in domestic dogs in Maun, Botswana, an area adjacent to the Okavango Delta, as a proxy for seroprevalence that would be expected in the wild canid and felid populations.

## Materials and methods

### Animals

The University of Tennessee Knoxville Institutional Animal Care and Used Committee approved this research proposal on March 27, 2015; #2333–0315. All blood collections were

done under the direct supervision of the veterinary members of the Maun Animal Welfare Society (MAWS). The majority (n = 128/233) of blood samples from domestic dogs were collected using a convenience sampling strategy at MAWS in Maun, Botswana, from dogs who were presented for castration and vaccination. The remaining blood samples (n = 105/233) were collected in surrounding villages. The uncastrated domestic dogs in this area were highly unlikely to have been vaccinated and be seropositive to the viral diseases due to vaccination cross-reaction because MAWS is the main veterinary clinic in the area for the low income population, and they will not vaccinate animals unless they are also castrated at the time of vaccination. Therefore, animals that were reproductively intact are likely to be unvaccinated. Blood samples were collected from a peripheral vein, transported on ice, and stored at -20˚C until testing.

### Sample analyses

Seroprevalence for CAV, CPV, and CDV were assessed using Biogal Titer Check (Biogal Galed Laboratories, Kibbutz Galed, Israel) following manufacturer's instructions. The Biogal Titer Check results were reported with a change of color and 'negative', 'positive', 'highly positive', or 'inconclusive' as the range of possible results. The vector-borne diseases, *Ehrlichia spp.*, *B. burgdorferi*, *Anaplasma spp.*, and *D. immitis*, were assessed with IDEXX 4DX SNAP ELISA (IDEXX, Westbrook, Maine, USA) following manufacturer's instructions. The IDEXX SNAP ELISA results were reported as either 'positive' or 'negative' by a color change.

### Statistical analysis

Crude and factor-specific seroprevalence proportions of *Ehrlichia spp.*, *B. burgdorferi*, *Anaplasma spp.*, *D. immitis*, CDV, CAV and CPV, as well as their 95% exact confidence intervals were computed. The factors considered were age, sex, and location. Associations between seroprevalence and each of the above factors were assessed using the Chi-square or Fishers Exact tests as appropriate. Significance was set at $\alpha = 0.05$ for all statistical tests. Logistic regression was used to identify adjusted predictors of seropositivity for each of the pathogens of interest.

## Results

### Animal demographics

A total of 233 dogs were tested (Table 1). Female dogs made up the majority (54.5%) of the sampled dogs. Dogs were sampled during the months of June and July 2015 with the majority (55.4%) of the dogs being sampled in July. The age group distribution was 70% adults and 30% juveniles. Samples were collected at seven locations in Maun, Botswana, and surrounding villages: MAWS (n = 128), Khumaga (n = 13), Boro (n = 18), Mathapane (n = 16), Shorobe (n = 18), Sexaxa (n = 8), and Etsha (n = 32).

### Crude seroprevalence

Of 232 individuals, 0% were seropositive for *B. burgdorferi*, 2.2% were seropositive for *Anaplasma spp.*, and 0.9% were seropositive for *D. immitis* (Table 2). Out of the 233 animals, 21.0% were seropositive for *Ehrlichia spp.*, 19.7% were seropositive for CAV, 70% were seropositive for CPV, and 46.8% were seropositive for CDV.

### Predictors of seropositivity

Based on the results of the logistic model, there were no statistically significant predictors (sex, age, month, or location) (P>0.05) for seropositivity of either *Anaplasma spp.* or *D. immitus* (Table 3).

**Table 1. Characteristics of dogs included in a seroprevalence assessment of prior exposure to common pathogens in Botswana, 2015.**

| Variable | Number | Percent | 95% Exact Binomial Confidence Interval |
|---|---|---|---|
| Sex | | | |
| Female | 127 | 54.5 | 47.9, 61.0 |
| Male | 106 | 45.5 | 3.90, 52.1 |
| Age Category | | | |
| Adult | 163 | 69.7 | 63.6, 75.8 |
| Juvenile | 70 | 30.0 | 24.2, 36.3 |
| Location | | | |
| Boro | 18 | 7.7 | 4.6, 11.9 |
| Etsha | 32 | 13.7 | 9.6, 18.8 |
| Khumaga | 13 | 5.6 | 3.0, 9.4 |
| MAWS | 128 | 54.9 | 48.3, 61.4 |
| Mathapane | 16 | 6.9 | 4.0, 10.9 |
| Sexaxa | 8 | 3.4 | 1.5, 6.7 |
| Shorobe | 18 | 7.7 | 4.6, 11.9 |
| Month | | | |
| June 2015 | 104 | 44.6 | 38.1, 51.0 |
| July 2015 | 129 | 55.4 | 48.7, 61.8 |

Although age of the dog and location had significant unadjusted associations with the odds of CAV seropositivity, only age (OR = 4.4; p<0.0003) was significant in the final analysis; implying that the odds of CAV seropositivity was 4.4 times higher in adults dogs than in juveniles (Table 4). Similarly, only age had a significant association in the final model for CPV with adults having 4.4 times higher odds of CPV seropositivity (OR 4.4; p<0.0001) than juveniles (Table 4).

Based on the results of the logistic model, CDV seropositivity was significantly associated with age (p<0.0001), month (p = 0.0002), and geographical location of sampling (p = 0.0437) (Table 5). Although both age of the dog and geographical location had significant unadjusted association with the odds of *Ehrlichia spp.* seropositivity (Table 5), when both were offered to the model in a multivariable analysis, neither was significant. Therefore, there was no final multivariable model for *Ehrlichia spp.*

Controlling for the other two factors in the model, the odds of CDV seropositivity is 12 times higher among adult dogs than the juveniles (Table 6). Similarly, the odds of the dogs having seropositive results for CDV were 7.8 times higher in June than in July (Table 6). With respect to geographical location, only Khumaga (p = 0.0014) and Shorobe (p = 0.0481) had significantly different odds of canine seropositivity from MAWS, with the odds of the dogs in

**Table 2. Crude seroprevalence of selected pathogens among domestic dogs in Botswana, 2015.**

| Pathogen | n | Number of seropositive samples | Percentage of seropositive samples | 95% Confidence Interval |
|---|---|---|---|---|
| *Anaplasma spp.* | 232[a] | 5 | 2.2 | 0.7, 5.0 |
| *B.burgdorferi* | 232[a] | 0 | 0 | 0, 1.6 |
| *D.immitis* | 232[a] | 2 | 0.9 | 0.1, 3.1 |
| CDV | 233 | 109 | 46.8 | 40.2, 53.4 |
| *Ehrlichia spp.* | 233 | 49 | 21.0 | 16.0, 26.8 |
| CAV | 233 | 46 | 19.7 | 14.8, 24.4 |
| CPV | 233 | 163 | 70.0 | 63.6, 75.8 |

[a] One record had missing information for results of *Anaplasma*, *B.burgdorferi* and *D.immitis*

**Table 3. Factor-specific seroprevalence of *Anaplasma spp.* and *D.immitis* among dogs in Botswana, 2015.**

| Pathogen | n | Number of seropositive samples | Percentage of seropositive samples | P-value |
|---|---|---|---|---|
| ***Anaplasma spp.*** | **232** | **5** | **2.2** | |
| *Sex* | | | | |
| *Female* | 126 | 4 | 3.2 | 0.379 |
| *Male* | 106 | 1 | 0.9 | |
| Age Category | | | | |
| *Adult* | 162 | 5 | 3.1 | 0.326 |
| *Juvenile* | 70 | 0 | 0 | |
| *Month* | | | | |
| *June* | 104 | 1 | 1.0 | 0.383 |
| *July* | 128 | 4 | 3.1 | |
| Location | | | | |
| Boro | 18 | 0 | 0 | 0.938 |
| Etsha | 32 | 0 | 0 | |
| Khumaga | 13 | 0 | 0 | |
| MAWS | 127 | 5 | 3.9 | |
| Mathapane | 16 | 0 | 0 | |
| Sexaxa | 8 | 0 | 0 | |
| Shorobe | 18 | 0 | 0 | |
| ***D.immitis*** | **232** | **2** | **0.9** | |
| *Sex* | | | | |
| *Female* | 126 | 2 | 1.6 | 0.502 |
| *Male* | 106 | 0 | 0 | |
| Age Category | | | | |
| *Adult* | 162 | 2 | 1.2 | 1.000 |
| *Juvenile* | 70 | 0 | 0 | |
| *Month* | | | | |
| *June* | 104 | 1 | 1.0 | 1.000 |
| *July* | 128 | 1 | 0.8 | |
| Location | | | | |
| Boro | 18 | 0 | 0 | 0.242 |
| Etsha | 32 | 0 | 0 | |
| Khumaga | 13 | 0 | 0 | |
| MAWS | 127 | 1 | 0.8 | |
| Mathapane | 16 | 0 | 0 | |
| Sexaxa | 8 | 1 | 12.5 | |
| Shorobe | 18 | 0 | 0 | |

Shorobe being 5.8 times higher than those of the reference group (MAWS) (Table 6). By contrast, the dogs in Khumaga had significantly lower odds (OR = 0.072) of testing seropositive to CDV than dogs in MAWS. The odds of being seropositive for CDV among dogs from the other locations were not significantly different from that of MAWS. Based on the Hosmer-Lemeshow Goodness-of-fit test, there is no evidence that the canine distemper model did not fit the data well (p = 0.4813).

## Discussion

Domestic animals can serve as disease reservoirs for wildlife. Wild animal populations, including various canid and felid species, have the potential to be infected by CDV, CPV, CAV, *D.*

**Table 4. Factor-specific seroprevalence of CAV and CPV among dogs in Botswana, 2015.**

| Pathogen | n | Number of seropositive samples | Percentage of seropositive samples | P-value |
|---|---|---|---|---|
| **CAV** | **233** | **46** | **19.7** | |
| *Sex* | | | | |
| *Female* | 127 | 20 | 15.6 | 0.101 |
| *Male* | 106 | 26 | 24.5 | |
| *Age Category* | | | | |
| *Adult* | 163 | 41 | 25.2 | 0.001 |
| *Juvenile* | 70 | 5 | 7.1 | |
| *Month* | | | | |
| *June* | 104 | 26 | 25.0 | 0.097 |
| *July* | 129 | 20 | 15.5 | |
| *Location* | | | | |
| *Boro* | 18 | 6 | 33.3 | 0.043 |
| Etsha | 32 | 5 | 15.6 | |
| Khumaga | 13 | 2 | 15.4 | |
| MAWS | 128 | 22 | 17.2 | |
| Mathapane | 16 | 8 | 50 | |
| Sexaxa | 8 | 0 | 0 | |
| Shorobe | 18 | 3 | 16.7 | |
| **CPV** | **233** | **163** | **70.0** | |
| *Sex* | | | | |
| *Female* | 127 | 92 | 72.4 | 0.392 |
| *Male* | 106 | 71 | 67.0 | |
| *Age Category* | | | | |
| *Adult* | 163 | 130 | 79.8 | <0.001 |
| *Juvenile* | 70 | 33 | 47.1 | |
| *Month* | | | | |
| *June* | 104 | 78 | 75.0 | 0.152 |
| *July* | 129 | 85 | 65.9 | |
| *Location* | | | | |
| *Boro* | 18 | 15 | 83.3 | 0.025 |
| Etsha | 32 | 16 | 50 | |
| Khumaga | 13 | 6 | 46.2 | |
| MAWS | 128 | 92 | 71.9 | |
| Mathapane | 16 | 12 | 75 | |
| Sexaxa | 8 | 6 | 75 | |
| Shorobe | 18 | 16 | 88.9 | |

*immitis*, *B. burgdorferi*, *Ehrlichia spp*. and *Anaplasma spp*., which are pathogens carried by the domestic animal population. Free-ranging cheetah (*Acinonyx jubatus*) in Namibia have been reported to have antibodies to CPV (though this test cross-reacts with feline panleukopenia virus) and CDV [15]. A captive breeding group of African wild dogs in Tanzania showed 94% mortality after infection with CDV [16], and another group of African wild dogs in Kenya had increasing disease-related mortality rates (from 21% to 50%) and CDV antibodies (from 1–4% to 76%) over a three-year period [3]. In Chobe National Park, Botswana, in 1996, a pack of twelve African wild dogs was reduced to two animals following an outbreak of CDV [17]. While the prevalence in the current study was low (0.9%), *D. immitis* can infect wildlife, with reports of *D. immitis* in a captive lion in Spain [18] and a captive black-footed cat in Florida

**Table 5. Factor-specific seroprevalence of CDV and *Ehrlichia spp.* among dogs in Botswana, 2015.**

| Pathogen | n | Number of seropositive samples | Percentage of seropositive samples | P-value |
|---|---|---|---|---|
| **CDV** | **233** | **109** | **46.8** | |
| *Sex* | | | | |
| *Female* | 127 | 62 | 48.8 | 0.512 |
| *Male* | 106 | 47 | 44.3 | |
| *Age Category* | | | | |
| *Adult* | 163 | 99 | 60.7 | <0.001 |
| *Juvenile* | 70 | 10 | 14.3 | |
| *Month* | | | | |
| *June* | 104 | 71 | 68.3 | <0.001 |
| *July* | 129 | 38 | 29.5 | |
| *Location* | | | | |
| Boro | 18 | 14 | 77.8 | <0.001 |
| Etsha | 32 | 6 | 18.8 | |
| Khumaga | 13 | 2 | 15.4 | |
| MAWS | 128 | 51 | 39.8 | |
| Mathapane | 16 | 13 | 81.3 | |
| Sexaxa | 8 | 6 | 75.0 | |
| Shorobe | 18 | 17 | 94.4 | |
| ***Ehrlichia spp.*** | **233** | **49** | **21.0** | |
| *Sex* | | | | |
| *Female* | 126 | 28 | 22.1 | 0.748 |
| *Male* | 106 | 21 | 19.8 | |
| *Age Category* | | | | |
| *Adult* | 163 | 40 | 24.5 | 0.054 |
| *Juvenile* | 70 | 9 | 12.9 | |
| *Month* | | | | |
| *June* | 104 | 22 | 21.2 | 1.000 |
| *July* | 129 | 27 | 20.9 | |
| *Location* | | | | |
| Boro | 18 | 3 | 16.7 | <0.001 |
| Etsha | 32 | 0 | 0 | |
| Khumaga | 13 | 0 | 0 | |
| MAWS | 128 | 40 | 31.3 | |
| Mathapane | 16 | 3 | 18.8 | |
| Sexaxa | 8 | 2 | 25 | |
| Shorobe | 18 | 1 | 7.6 | |

[19]. Butler, *et al.* [20] indicated that domestic dogs in northwest Zimbabwe are a source of disease transmission for leopards (*Panthera pardus*), lions, and spotted hyenas (*Crocuta crocuta*), as these predator species feed on domestic dogs as prey. A survey of domestic dogs and African wild dogs in Kenya from 2001 to 2009, showed 16% of African wild dogs and 48% of domestic dogs had been exposed to CDV, 25% of African wild dogs and 64% of domestic dogs had been exposed to CPV, and 80% of African wild dogs and 86% of domestic dogs had been exposed to *Ehrlichia canis* [21]. Similar to the Kenya evaluation, the results of the present study revealed 46.8% CDV seropositititivity and 70.0% CPV seroposity in domestic dogs. However, it was found that lower seropositivity rates for *Ehrlichia spp.* (21.0%) were present as compared to the Kenya study.

**Table 6. Results of multivariable logistic regression showing predictors of CDV sero-positivity among domestic dogs in Botswana, 2019.**

| Predictor | Odds Ratio | 95% Confidence Interval | P-value |
|---|---|---|---|
| **Age** | | | |
| Adult | 12.4 | 4.8, 32.1 | <0.0001 |
| Juvenile | Referent | Referent | |
| **Month** | | | |
| June | 7.8 | 2.6, 23.1 | 0.0002 |
| July | Referent | Referent | |
| **Location** | | | |
| Boro | 0.7 | 0.2, 3.3 | 0.8022 |
| Etsha | 0.7 | 0.2, 1.9 | 0.7233 |
| Khumaga | 0.07 | 0.01, 0.5 | 0.0014 |
| Mathapane | 1.2 | 0.2, 6.5 | 0.5889 |
| Sexaxa | 1.2 | 0.2, 9.6 | 0.6601 |
| Shorobe | 5.8 | 0.6, 57.8 | 0.0481 |
| MAWS | Referent | Referent | |

When comparing regional infectious disease prevalence differences, similar seroprevalence for viral diseases in domestic dogs were determined in Botswana as was reported in a similar study performed in Zimbabwe in 2012 [6]. A study of domestic dogs in northwest Zimbabwe reported that 34% had antibodies to CDV, 84% had antibodies to CPV, and 13% had antibodies for CAV [6]. These results are similar to those of the present study of 46.8% for CDV, 70.0% for CPV, and 19.7% for CAV. Another seroprevalence study evaluating domestic dogs in northeast Namibia in 1993 and 1994 found similar exposure to CDV (44.3%), but lower prevalence of CPV (47.1%) and higher prevalence of CAV (64.3%) [22]. Both Zimbabwe and Namibia border Botswana on its eastern and western edges, respectively. Viral diseases spread from animal to animal so localized differences in exposure are expected, but vectored pathogens depend on prevalence of the pathogen, prevalence of the vector, and on contact between the vector and the susceptible animal host. Williams, *et al.* [23] assessed prevalence of several hemoparasites, including *Ehrlichia spp.*, in domestic dogs, lions, spotted hyena, and African wild dogs in 2009 to 2011 in Zambia, another country in southern Africa that borders Botswana, and samples were evaluated by polymerase chain reaction which only reveals active infections rather than current and past exposure. No carnivores had positive results for *E. canis* or *E. ewingii* [23], which does not rule out presence of *Ehrlichia* in the study area.

As ticks are the vectors for *B. burgdorferi*, *Ehrlichia spp.* and *Anaplasma spp*, tick prevalence is a crucial factor in the spread of these pathogens. While no recent studies have reported tick prevalence in northern Botswana, Eygelaar, *et al.* [24] reported that African buffalo (*Syncerus caffer*) in the Okavango Delta had a lower prevalence of tick-borne diseases than African buffalo in Chobe National Park, a region in northeast Botswana that is closer to the study region assessed by McRee, *et al.* [6]. Perhaps the same cause is responsible for the reduced tick-vectored diseases in African buffalo and domestic dogs in the Okavango Delta compared to northwest Zimbabwe. Eygelaar, *et al.* [24] hypothesized that veterinary fences in the Okavango Delta prevented direct contact between the African buffalo and cattle, which were not present in Chobe National Park. Perhaps these same fences reduce tick spread from wildlife to domestic dogs and vice versa, which limits the spread of the vector-borne diseases in the Okavango Delta which were not present in northwest Zimbabwe. More research must be performed to determine if the differences in *Ehrlichia spp.* seroprevalence is due to a reduction in total tick numbers or another cause, such as physical barriers. The high viral disease seroprevalence,

similar to those in northwest Zimbabwe, is likely due to lack of vaccination. While strong efforts are being actively put forth by local not-for-profit organizations, the number of unvaccinated dogs remains much greater than the number of vaccinated dogs.

Factor-specific seroprevalence indicated significant associations between seropositivity to these common canine infectious diseases and age, month, and location. Adults are more likely to be seropositive for CAV, CDV, and CPV, which is likely due to having more time to be exposed to the viruses than juveniles. There was a significant association between seropositivity for CDV and month with June, having higher risk than July. Lastly, geographical location is a risk factor for CAV, CDV, and CPV because viral pathogens are transmitted either by direct contact or contact with bodily fluids. Therefore, geographical locations with high rates of these pathogens allow easy transmission to naïve individuals. Location was also a risk factor for *E. canis* perhaps because of tick concentrations in certain locations or due to an increase in the pathogen in the dogs of certain locations that perpetuates the elevated infection rate (ticks can spread the pathogen transstadially, but not transovarially [7]).

While the viral pathogens evaluated in this study cannot infect humans, some of the vectored pathogens can affect humans. In addition to affecting domestic and wild animal populations, *E. canis* and *E. ewingii* have been reported in humans [25, 26]. *Borrelia burgdorferi*, the causative organism for Lyme disease, and *A. phagocytophilum* [27, 28] are also zoonotic. The 'One Health' paradigm, a collaborative approach to animal, human, and environmental health that recognizes their interconnectivity, is particularly important, since reducing disease risk in domestic animals will reduce disease risk in wildlife and human populations. By increasing vaccination and reducing tick burden in domestic dogs, human health and environmental health, in the case of wildlife, are improved [29, 30].

One of the limitations of this study was that serology detects exposure to the pathogen, but it does not determine the rate of active infections. Thus, seroprevalence indicates that pathogen exposure has occurred, but current risk of infection is unknown. This aspect is important for human and wildlife health because new infections may increase disease burden in these populations.

Information regarding disease prevalence is necessary to determine domestic animal, wildlife, and human disease risk. This study reveals the need for local tick surveys to determine the cause of tick-borne pathogen prevalence differences between Botswana and surrounding countries. In conclusion, further disease testing and vaccination of both domestic dogs and wildlife would benefit domestic animals, wildlife, and humans in the Okavango Delta region of Botswana.

## Supporting information

**S1 File. Data analyzed including ELISA results for 233 blood samples collected from dogs in northern Botswana.**
(XLS)

## Acknowledgments

We would like to thank all of the staff at the Maun Animal Welfare Society, including Tana, Justine, KC, and Nation. We would also like to thank Hansje de Waard, Deirdre Halloran, and Anna Hewitt for generous assistance in sample collection.

## Author Contributions

**Conceptualization:** Hayley Adams, Melissa Kennedy.

**Data curation:** Riley Thompson, Melissa Kennedy.

**Formal analysis:** Agricola Odoi, Melissa Kennedy.

**Funding acquisition:** Riley Thompson, Melissa Kennedy.

**Investigation:** Melissa Kennedy.

**Methodology:** Riley Thompson, Agricola Odoi, Melissa Kennedy.

**Project administration:** Melissa Kennedy.

**Resources:** Hayley Adams.

**Supervision:** Melissa Kennedy.

**Validation:** Agricola Odoi.

**Writing – original draft:** Riley Thompson.

**Writing – review & editing:** Hayley Adams, Agricola Odoi, Melissa Kennedy.

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
