## [Decision Letter · Decision Letter 0]

30 Aug 2019

PONE-D-19-20072

Seroprevalence of viral and vector-borne bacterial pathogens in domestic dogs (Canis familiaris) in northern Botswana

PLOS ONE

Dear Melissa Kennedy

Thank you for submitting your manuscript to PLOS ONE. After careful consideration, we feel that it has merit but does not fully meet PLOS ONE’s publication criteria as it currently stands. Therefore, we invite you to submit a revised version of the manuscript that addresses the points raised during the review process.

ACADEMIC EDITOR: 

Many thanks for submitting your manuscript to PLOS One

Your manuscript was reviewed by two reviewers who made some good useful suggestions to improve it

Please write a detailed response to reviewers when modifying your document to make their lives a bit easier.

I look forward to reading the modified manuscript

Best of luck with your modifications

Thanks

Simon

We would appreciate receiving your revised manuscript by Oct 14 2019 11:59PM. To enhance the reproducibility of your results, we recommend that if applicable you deposit your laboratory protocols in protocols.io, where a protocol can be assigned its own identifier (DOI) such that it can be cited independently in the future. For instructions see: http://journals.plos.org/plosone/s/submission-guidelines#loc-laboratory-protocols

We look forward to receiving your revised manuscript.

Kind regards,

Simon Russell Clegg, PhD

Academic Editor

PLOS ONE

Journal Requirements:

3. In your Methods section, please provide additional details regarding participant consent from the animals owners. In the ethics statement in the Methods and online submission information, please ensure that you have specified (1) whether consent was informed and (2) what type you obtained (for instance, written or verbal). If the need for consent was waived by the ethics committee, please include this information.

4. In your Methods, please state where the participants were recruited for your study.

5. In your Methods, please state the volume of the blood samples collected for use in your study.

Reviewers' comments:

Reviewer's Responses to Questions

**Comments to the Author**

1. Is the manuscript technically sound, and do the data support the conclusions?

Reviewer #1: Yes

Reviewer #2: No

2. Has the statistical analysis been performed appropriately and rigorously? 

Reviewer #1: Yes

Reviewer #2: Yes

3. Have the authors made all data underlying the findings in their manuscript fully available?

Reviewer #1: Yes

Reviewer #2: Yes

4. Is the manuscript presented in an intelligible fashion and written in standard English?

Reviewer #1: Yes

Reviewer #2: Yes

5. Review Comments to the Author

Reviewer #1: Dear Editor

I have read through the MS entitled ’Seroprevalence of viral and vector-borne bacterial pathogens in domestic dogs (Canis familiaris) in northern Botswana’ sent to me for review. The MS is technically sound and the information contributes to the understanding of disease dynamics in domestic dog population in northern Botswana and by implication the disease condition in the wild felid and canid populations. However, there are some aspect of the study that needs to be addressed in order to enhance the clarity of the paper and improve on its quality in general. My comments to the authors is attached

I recommend moderate revision of the manuscript before it can accepted for publication.

Thank you

Joshua Kamani

Reviewer #2: Ms. No. PONE-D-19-20072

Seroprevalence of viral and vector-borne bacterial pathogens in domestic dogs (Canis

familiaris) in northern Botswana

Plos one

This study is a survey on the main viral, bacterial and parasitic pathogens in domestic dogs living in an area that has been little investigated such as the Botswana.

It is interesting to use domestic dogs as sentinels for infectious disease exposure in wildlife.

However, this study's objectives are not clearly stated and the methods are incomplete and, at times, difficult to follow.

The epidemiologic study was completely descriptive.

In addition, sampling methods are unclear.

It would be optimal to better specify the age of the dogs examined and how age was determined.

The withdrawal period is so short that I don't think it can really influence the positivity differences. I would eliminate it.

The discussion should be re-evaluated with greater attention and the hypotheses on the role of ticks in the transmission of some pathogens rewritten, please.

I would reduce the tables to only one.

Tables 1 and 2 show 232 dogs analyzed while 3 and 4 show 233 dogs, please check the tables.

References

Check the bibliographic references

Insert in italic the different pathogens

6. PLOS authors have the option to publish the peer review history of their article (what does this mean?). If published, this will include your full peer review and any attached files.

Reviewer #1: Yes: Joshua Kamani

Reviewer #2: No

---

## [Author Response · Author response to Decision Letter 0]

15 Oct 2019

Reviewers’ comments:

Laboratory methods – given that the only techniques used involved ELISA kits per manufacturers’ recommendation, none were deposited.

The style requirements were reviewed.

Owners’ verbal consent for participation, the locale of participants, and the volume of blood collected were addressed in the text.

Reviewer #1:

The short title was modified per the reviewer’s recommendations.

The italics for “spp.” designations were removed.

“total of” was removed from the sentence in line 37.

The issue of tick infestation and control were addressed in the text

Lines 60-61 and line 64 were modified per the reviewer’s recommendations.

The tick vector for each organism was included.

Rephrasing of lines 87-89 was made.

As stated in the manuscript, rabies was not assessed because of unknown vaccination status for rabies, and the lack of a point-of-care assay for this assessment.

The various aspects of demographic data were addressed in the text.

The use of whole blood was addressed

The various properties of the kits were addressed.

Color change evaluation was addressed.

Tables were condensed. 

Lines 202-206 were modified per the reviewer’s recommendation.

The reference corrections were made.

Reviewer #2:

The hypothesis, objectives and methods were clarified.

Demographic data was addressed. 

The discussion section was adjusted. 

Tables were reduced: 

Reviewer Comment

Line 134: is it 232 or 233?

Response

The total sample size was 233. However, there was one missing data point for test results of the following pathogens: Anaplasma spp., B.burgdorferi and D.immitis. Due to that missing data, these pathogens had 232 instead of 233. The rest of the pathogens did not have missing data and therefore had a total of 233. These explanations have been added as footnotes on tables 2, 3a and 3b 

Reviewer Comment

Table 2: I am not sure the authors explained why there are two populations; 232 233 being tested for different pathogens.

Response

There is only one population of dogs. However, as explained above there was there was one missing data point for test results for Anaplasma spp., B.burgdorferi and D.immitis. Therefore these pathogens had data for only 232 dogs instead of 233 that the rest of the pathogen had.

Reviewer Comment

Table 3: The table can be presented in a better format without having to repeat the variables in columns 1 and 2 for the two pathogens. Why are some variables in column 1 in italics? (see throughout the tables) 

Response

We thank the reviewer for their suggestion on improving the presentation of our results. As per the suggestion, tables 3, 4 and 5 have now been combined into Table 3. However, since the new Table 3 cannot fit on one page, it has been split into Table 3a and Table 3b. This change has allowed us to stop repeating information from column 1. However, we have retained column 2 because the data changes between pathogens depending on whether there was missing data or not. 

Reviewer Comment

Tables 4 and 5 should be merged similar to comments made for table 3 to avoid duplicating the variables in columns 1 and 2.

Response:

As stated above tables 3, 4 and 5 have not been combined into Table 3. Since the Table 3 cannot fit on one page, it has been split into Table 3a and Table 3b. This change has allowed us to stop repeating information from column 1. However, we have retained column 2 because the data changes between pathogens depending on whether there was missing data or not. 

Reviewer Comment

Line 240-41: What could influence the significant association of CDV with June than July? The study duration of 2 months to me is too short to make any meaningful deduction other than chance findings.

Response:

We thank the reviewer for noting this association and seeking an explanation. Unfortunately, this being a sero-survey, the scope of the study was limited and so we were not able to investigate this association further. However, due to the very strong association (OR = 7.8; p=0.002), it is unlikely to be due to chance. Suffice it to say that future studies will need to investigate this association further. We have added this comment in the manuscript.

Reference corrections were made.

---

## [Decision Letter · Decision Letter 1]

19 Nov 2019

PONE-D-19-20072R1

Seroprevalence of viral and vector-borne bacterial pathogens in domestic dogs (Canis familiaris) in northern Botswana

PLOS ONE

Dear Melissa Kennedy

Thank you for submitting your manuscript to PLOS ONE. After careful consideration, we feel that it has merit but does not fully meet PLOS ONE’s publication criteria as it currently stands. Therefore, we invite you to submit a revised version of the manuscript that addresses the points raised during the review process. Please note that the modifications here are very minor.

Many thanks for resubmitting your manuscript to PLOS One

It has again been reviewed by two expert reviewers, and they have come back with some very minor typographical and grammatical comments

If you can address these comments, then I can recommend the article for publication

Please do not feel that you need to write a full rebuttal to reviewers comments. Merely a line saying that all comments were addressed, and a comment on which ones were not and why.

Wishing you the best of luck with your minor revisions

Thanks

Simon

We would appreciate receiving your revised manuscript by Jan 03 2020 11:59PM. To enhance the reproducibility of your results, we recommend that if applicable you deposit your laboratory protocols in protocols.io, where a protocol can be assigned its own identifier (DOI) such that it can be cited independently in the future. For instructions see: http://journals.plos.org/plosone/s/submission-guidelines#loc-laboratory-protocols

A marked-up copy of your manuscript that highlights changes made to the original version. This file should be uploaded as separate file and labeled 'Revised Manuscript with Track Changes'.An unmarked version of your revised paper without tracked changes. This file should be uploaded as separate file and labeled 'Manuscript'.

We look forward to receiving your revised manuscript.

Kind regards,

Simon Russell Clegg, PhD

Academic Editor

PLOS ONE

Reviewers' comments:

Reviewer's Responses to Questions

**Comments to the Author**

1. If the authors have adequately addressed your comments raised in a previous round of review and you feel that this manuscript is now acceptable for publication, you may indicate that here to bypass the “Comments to the Author” section, enter your conflict of interest statement in the “Confidential to Editor” section, and submit your "Accept" recommendation.

Reviewer #1: (No Response)

Reviewer #3: (No Response)

2. Is the manuscript technically sound, and do the data support the conclusions?

Reviewer #1: (No Response)

Reviewer #3: Yes

3. Has the statistical analysis been performed appropriately and rigorously? 

Reviewer #1: (No Response)

Reviewer #3: Yes

4. Have the authors made all data underlying the findings in their manuscript fully available?

Reviewer #1: (No Response)

Reviewer #3: Yes

5. Is the manuscript presented in an intelligible fashion and written in standard English?

Reviewer #1: (No Response)

Reviewer #3: Yes

6. Review Comments to the Author

Reviewer #1: (No Response)

Reviewer #3: I found this manuscript very interesting, and current, particularly with the sad plight of the African wild dog. It is an interesting way of looking at potential disease risk to these animals.

As this is a second submission, I have made mainly minor comments to the manuscript as I think it is well written and interesting.

Line 36- numbers under 10 should be written in full

Line 63- et al., should be in italics throughout as it is Latin

Line 70- would serious illness and/or death maybe be better as morbidity and mortality?

Line 80- would cross species transmission sound better?

Line 105- following manufacturers instructions may sound better, or as per manufacturers instructions

Line 108- you can shorten B. burgdorferi and D. immitis as you have already mentioned them in full. The only time it needs to be in full after the first time is at the start of a sentence. Please modify throughout

Line 114- comma after CPV

Line 125- you mention age distribution, but there is 0.3% missing. Where does that animal fit?

Tables may look better center aligned

Table 2. Why is there a difference in number tested for Anaplasma, Borrelia and Dinofilaria compared to the others? I don’t remember reading that in the text anywhere

Line 147- in the final what? It didn’t make sense to me

Line 153- comma after (Table 5)

Line 161- comma after MAWS

Table 3, 4 and 5- Is it possible to fill in some of the gaps (even a hyphen may help), or merge cells?

Line 187-190- this didn’t quite read correctly, but I cant work out how to reword it. Maybe have a look at it and see if you could modify it?

Line 196- wild doesn’t need capitalising

Line 197- comma after 2009

Line 199 – I think this is the first mention of E. canis so it needs to be in full

Line 201- Avoid using us, we, our etc in scientific writing

Line 203- change was to were

Line 218- E. ewingii needs to be in full as it’s the first mention of it

Line 219- Ehrlichia needs capitalising

Line 223- had a lower prevalence

Line 241- comma after month

Line 250- B. burgdorferi needs to be in full as it’s the start of a sentence

Reference 10- names don’t need capitalising

7. PLOS authors have the option to publish the peer review history of their article (what does this mean?). If published, this will include your full peer review and any attached files.

Reviewer #1: None

Reviewer #3: No

---

## [Author Response · Author response to Decision Letter 1]

25 Nov 2019

Minor revisions were addressed.

---

## [Editor Report · Decision Letter 2]

23 Dec 2019

Seroprevalence of viral and vector-borne bacterial pathogens in domestic dogs (Canis familiaris) in northern Botswana

PONE-D-19-20072R2

Dear Dr.Kennedy

We are pleased to inform you that your manuscript has been judged scientifically suitable for publication and will be formally accepted for publication once it complies with all outstanding technical requirements.

With kind regards,

Simon Russell Clegg, PhD

Academic Editor

PLOS ONE

Additional Editor Comments (optional):

Many thanks for submitting your revised manuscript to PLOS One

As you have addressed all the comments (thank you) I have recommended the manuscript for publication

I wish you all the best for your future research

Many thanks

Simon
---

## [Editor Report · Acceptance letter]

30 Dec 2019

PONE-D-19-20072R2 

Seroprevalence of viral and vector-borne bacterial pathogens in domestic dogs (Canis familiaris) in northern Botswana 

Dear Dr. Kennedy:

I am pleased to inform you that your manuscript has been deemed suitable for publication in PLOS ONE. Congratulations! Your manuscript is now with our production department. 

With kind regards,

on behalf of

Dr. Simon Russell Clegg 

Academic Editor

PLOS ONE